# The Need for Trauma Management Training and Evaluation on a Prehospital Setting

**DOI:** 10.3390/ijerph192013188

**Published:** 2022-10-13

**Authors:** Blanca Larraga-García, Manuel Quintana-Díaz, Álvaro Gutiérrez

**Affiliations:** 1Escuela Técnica Superior de Ingenieros de Telecomunicación, Universidad Politécnica de Madrid, 28040 Madrid, Spain; 2Hospital La Paz Institute for Health Research, IdiPAZ, 28029 Madrid, Spain

**Keywords:** trauma, medical education, training, evaluation criteria, protocol, prehospital

## Abstract

Trauma is one of the leading causes of death in the world, being the main cause of death in people under 45 years old. The epidemiology of these deaths shows an important peak during the first hour after a traumatic event. Therefore, learning how to manage traumatic injuries in a prehospital setting is of great importance. Medical students from Universidad Autónoma performed 66 different simulations to stabilize a trauma patient on a prehospital scene by using a web-based trauma simulator. Then, a panel of trauma experts evaluated the simulations performed, observing that, on average, an important number of simulations were scored below 5, being the score values provided from 0, minimum, to 10, maximum. Therefore, the first need detected is the need to further train prehospital trauma management in undergraduate education. Additionally, a deeper analysis of the scores provided by the experts was performed. It showed a great dispersion in the scores provided by the different trauma experts per simulation. Therefore, a second need is identified, the need to develop a system to objectively evaluate trauma management.

## 1. Introduction

Trauma is one of the main causes of death in the world, being the leading one for people under 45 years old [1]. These injuries result from traffic accidents, drowning, falls or violence, causing millions of deaths around the globe. The trauma death distribution has been classified, over the years, as a trimodal distribution, identifying three important peaks [2]. Nonetheless, since the 2000s, this distribution has changed. The new distribution identifies two peaks as the third one is not detected [3,4,5]. Nevertheless, immediate deaths are still an important number, whereas the second and the third peaks merge, declining the number of deaths over time [6,7]. As these immediate deaths are still important, training in how to deal with trauma injuries is of great importance. In 1978, The Advanced Trauma Life Support (ATLS) was created by the American College of Surgeons. Since that year, it has been taught all around the world, becoming a trauma standard in more than 44 countries and being delivered in over 60 countries [8]. Nevertheless, this training is designed for primary care physicians, surgeons, emergency medicine physicians and physician extenders involved and not for medical students. Due to this fact, medical students are requesting to receive more trauma courses [9,10], and the American College of Surgeons has created another trauma course named Trauma Evaluation and Management (TEAM), which is an introductory trauma training [8]. Additionally, the ATLS training has a specific number of students per year that cannot be surpassed. Therefore, there is a waiting list to take this training which will take, depending on the country, from 6 months to 2.5 years to complete the ATLS training. As a consequence, new trauma training is being created [11,12,13,14,15,16,17,18,19].

In 1981, just after the release of the ATLS training, Prehospital Trauma Life Support (PHTLS) started [20]. Even though this training follows the same principles of the ATLS, it is specifically designed for the prehospital environment and personnel. The PHTLS training focuses on the principles of prehospital attention, which are called the Golden Principles of Prehospital Trauma Care [21]. PHTLS provides an understanding of the anatomy and physiology, the pathophysiology of trauma and the assessment and healthcare of the trauma patient using the ABCDE approach. The last version of the PHTLS introduces a new approach which is called XABCDE. In this approach, the hemorrhage, X, is recognized as a potentially irreversible cause of death and, therefore, an immediate threat to take care of. Therefore, having the X prior to the ABCDE approach sets the necessity to take care immediately of any external hemorrhage. This should be done just right after ensuring the safety of the prehospital personnel and the patient and before assessing the airway.

A severe hemorrhage, in particular arterial bleeding, has the potential to cause a complete loss of blood volume in a relatively short time. Depending on the bleeding, this could happen in only some minutes. As in the prehospital setting, where it is not possible to transfuse blood, it is extremely important that the hemorrhage is controlled and that this blood loss does not happen before getting to the healthcare facility.

Therefore, trauma management education is key to teaching and learning how to better treat trauma patients. According to Jouda and Finn [10], trauma management training in undergraduate education is poorly described within the literature. Additionally, when the publications are found, there is a lack of common structure on the training developed. In some faculties, the time devoted to trauma management training is less than 10 h, and in others, more than 2 months [22]. Moreover, the resources used in teaching trauma management are also diverse, from theoretical lessons and discussion groups to some skill stations [9,22,23]. Trauma Management Courses (TMCs) are sometimes implemented in undergraduate medical education [24,25,26]. Their model is more structured, and they use different types of simulation modalities to support trauma management training, from technical skill stations to high-fidelity mannequins. The use of simulation in trauma management is of great importance as it is quite difficult to have hands-on experience when dealing with a real trauma patient. The time to stabilize a trauma patient is usually limited, and therefore, patient safety is prioritized versus teaching in that context. Additionally, these TMCs evaluate the improvement achieved once it is finished. This is done by using multiple-choice questionnaires or by analyzing the simulation performed with the trauma training instructors. Borggreve et al. propose that it will be effective in improving trauma management skills to incorporate TMCs into the medical curriculum [27]. Nevertheless, there is evidence that postgraduate trauma education has an impact on the performance of the student. Some studies show clear improvements in trauma management after following trauma guidelines, clinical pathways, or protocols [28,29,30,31,32,33,34,35]; therefore, defining all these methods to correctly teach and evaluate them is of great importance. These are three levels of standard procedures, which go from more general guidelines, to more precise, which are first clinical pathways and then protocols [36,37,38]. Additionally, to study deviations from protocols is a clear need to be able to objectively evaluate knowledge acquisition [39]. This may allow us to study the adequacy of the protocol, incorporate flexibility in the protocols and constantly evaluate whether changes might be considered in the analyzed protocol [40,41,42,43,44,45,46]. Since the ATLS and the PHTLS courses were developed, a general framework was provided with respect to how to manage trauma injuries. Nevertheless, attrition and low compliance rate with that framework are important issues even in major trauma facilities [47,48]. The attrition rate is affected by several aspects, such as the time from the trauma training and the volume of trauma patients treated regularly. Therefore, trauma training needs to be repeated, especially if the volume of trauma patients treated regularly is not high; if not, the attrition rate diminishes. The studies that investigate this aspect evaluate trauma adherence to protocols by using exams and practical skills stations by the Objective Structured Clinical Examination (OSCE) performance. These skill stations are evaluated by trainers using a checklist which could be improved by incorporating objective information directly from the simulation modality used within the skill station. However, the low compliance rate is not deeply investigated, and only a few studies focus on this aspect [49,50,51].

These findings have raised concerns with respect to the current prehospital trauma management education, as well as how to objectively evaluate knowledge acquisition in trauma management. Therefore, the purpose of this investigation is to study how prehospital trauma management cases are handled and how they are evaluated in undergraduate medical education. To accomplish this goal, the trauma management performance of medical students is analyzed.

## 2. Materials and Methods

### 2.1. Study Design

Different trauma scenarios were created based on two different trauma injuries: pelvic and lower limb lesions occurring in a prehospital setting. Therefore, two different scenarios were trained: Scenario 1: a prehospital pelvic trauma scenario; and Scenario 2: a prehospital lower limb trauma scenario. Considering these scenarios, 8 different trauma case studies were generated. All of them were included in a web-based trauma simulator (https://github.com/Robolabo/trauma-simulator, accessed on 10 October 2022) which was used to accomplish the trauma management scenarios. In Figure 1, the main page of the simulator is shown in which the virtual patient, some of the actions that could be accomplished, and the vital signs of the patient are displayed. The scenarios were defined by the trauma experts in which the sex of the patient, age, and part of the body affected, together with the vital signs of the patient at the time of the trauma, are provided, as shown in Figure 2.

Additionally, it is possible to interact with the patient, and a remaining lifetime is provided. It creates a more realistic scenario in which a fast and efficient response should be provided under stressful circumstances. The treatment of the virtual patient is done individually; therefore, every participant must decide on the treatment of the patient. Once all the trainees performed the simulations, a panel of trauma experts composed of 18 trauma clinicians evaluated the trauma management performance of the trainees. All the experts are consultants of the intensive care medicine department of different hospitals throughout Spain. All of them are members of the Spanish Society of Intensive Care Medicine and Coronary Units (SEMICYUC) and with at least 3 years’ experience after residency education in Intensive Care Medicine. The evaluation of the simulations was done once the simulations were finished. The data was extracted automatically from the web-based simulator so that the experts simply receive the simulated trauma scenario, the simulation number, the sequence of actions accomplished by the trainee and the evolution of the virtual trauma patient. No information about the trainees was provided to the trauma experts.

### 2.2. Participants

The trainees were final-year medical students from the Universidad Autónoma de Madrid with previous training in trauma management. They were students who voluntarily agreed to participate in the study. They were introduced to the web-based simulator in an online session. After the introduction session, a user manual, together with a demo video about the simulator’s use, was provided to the participants. The participants performed 66 simulations in total.

### 2.3. Data Analysis

All the data generated along the simulations were gathered automatically into the simulator database: all the actions taken, the moment in time in which the actions were accomplished, and the impact on the vital signs of the virtual patient. Therefore, this is automatically obtained from the database associated with the web-based trauma simulator to evaluate the performance of the trainees. As several aspects should be considered within the evaluation, such as the order in which the actions were taken, if the actions accomplished were adequate or not, the time taken along the treatment or vital signs evolution of the patient, the data was processed in Python.

Then, the panel of experts scored the 66 simulations performed by trainees, considering all the data provided by the simulator. The measurement unit used in this study was a global score from 0 to 10. Being 0 the minimum score, which means that the simulation performed did not comply at all with the treatments that should be provided to a trauma patient, and 10 the maximum score, which means that the simulation reflected perfectly the treatment that should be provided to a trauma patient. This score is provided based on the sequence of actions accomplished during the simulation. Therefore, the score compares the sequence of actions taken by the trainee with the sequence of actions that the experts would have applied for the same trauma scenario. Then, all the scores provided by the experts are analyzed per simulation, studying the average and the standard deviation values as shown in Table 1. Additionally, a deeper study is done by analyzing the whole range of score values provided by all the trauma experts.

## 3. Results

The average results obtained for all the simulations show that an important number of simulations received an average score between 3.5–4.5, as shown in Figure 3. In fact, the average value is 4.17 ± 1.09, quite similar to the median value (4.16 ± 1.23) and the mode (3.80).

Consequently, all the scores provided by the panel of experts were further analyzed per trauma Scenario. In Table 1, all the averages and standard deviations of the scores for all the simulations are shown. In the first column, the first trauma scenario values are shown, and in the second one, the second trauma scenario scores. In the end, the average score and standard deviation per trauma scenario are highlighted. The results are similar per trauma scenario, being a little bit higher for the pelvic prehospital trauma scenario. The Wilcoxon Rank-Sum test has been used in python to compare the two samples: the pelvic prehospital trauma scenario and the lower limb one. This test examines whether the populations differ in the median, the null hypothesis being that both samples are identical with a significance level of *p* < 0.05. Nevertheless, no statistical difference is perceived.

In order to further analyze all the score values provided per simulation, a boxplot analysis was performed. The results in Figure 4 show all the individual scores provided by the experts for Scenario 1: 36 different simulations were analyzed by the 18 experts obtaining a quite diverse response. Having a look at Figure 4, it is perceived that for some simulations, there are completely different evaluations. For example, in simulation 17, one expert provided a score of 1, and another expert provided a score of 9. This could be considered exceptional, but, in general, an important dispersion on all the scores provided is perceived. The differences between the maximum and the minimum scores are quite important. This difference has an average value of 5.68 and a median value of 6. In fact, looking at Figure 4, only five simulations had a difference between the maximum and the minimum score, which was lower than 5. This means that, for the simulations of this trauma scenario, experts provided quite diverse responses.

In Figure 5, the scores provided for all the simulations of Scenario 2 are shown: 30 different simulations were analyzed. Once again, there are simulations with an important dispersion on the score values provided by the experts, such as simulations 17 and 25. For simulation 17, a score of 1 was provided by one expert and a score of 8 by another one. For simulation 25, a minimum score of 0 was obtained, whereas a maximum score of 7 was given by another expert. Once again, upon analyzing the differences between the maximum and minimum scores provided per simulation, they are quite important. The average value of these differences for all the simulations is 5.32, whereas the median value is 5.53. In this case, seven simulations show a difference between the maximum and minimum below 5; this is more than in the previous trauma scenario, but it only represents 23% of all the simulations. Furthermore, obtaining a difference of a score higher than 5 is quite an important difference in an evaluation score.

## 4. Discussion

The average scores obtained show that trainees do not comply with what is expected to do when treating a trauma patient, considering the actions taken for the stabilization of a trauma patient. The average scores obtained in the work presented in this article are similar to the trauma management scores obtained in [25,26]. In these previous studies, the average score obtained was 5.7 out of 10 for the first study [25] and 4.5 for the second one [26]. Therefore, this confirms the need to train in trauma management skills too. Nevertheless, this article presents the first study found in the literature in which the evaluation is obtained based on all actions performed by trainees and recorded directly by the simulator.

In this pilot study, not only the need to train prehospital trauma management is detected. Additionally, it is clear that different criteria are followed by experts when evaluating a prehospital trauma scenario. Consequently, there is a need to develop an objective evaluation system for trauma management. To do so, it is important to understand which are the main principles behind prehospital trauma management and the best way to learn and teach them. The number of actions to accomplish a trauma management scenario is large, and therefore, it is important to be able to prioritize actions and do so quickly. Moreover, understanding in detail what are the implications that each of the actions has on the vital signs of the patient is key. Additionally, identifying the actions to apply according to the trauma lesion suffered and the resources available is another important aspect to analyze in detail before developing an automated evaluation system. In some studies [52,53], an evaluation procedure is developed, providing different scores for different skills and combining them in questionnaires that either trainees or trainers have to fulfill after a simulation. Additionally, electronic surveys have been developed, as in Harrington et al. [13], and some scales, such as the Trauma-NOTECHS, have been used [54]. Nevertheless, the evaluation performed within this study takes the actions accomplished directly from the simulator, with the right order and the moment in which they were accomplished. This sets an important basis for developing a more comprehensive evaluation method, as the data evaluated are purely objective.

On one side, the need to further train prehospital trauma management is clear from the scores provided by the experts in which, on average, a small number of simulations comply with the treatment that should be provided to a trauma patient. This shows a gap in the prehospital trauma education landscape in undergraduate education. Moreover, according to Jouda and Finn [10], the best simulation method and procedure to teach trauma management to medical students has not yet been established. This is a field that needs further development [55,56]. Additionally, prehospital settings should be further trained as doctors should be able to attend to a trauma patient as soon as possible, with the knowledge of the available resources depending on the environment in which they need to support the patient. Even though there has been an increase in the number of prehospital trauma management training, this needs to be further improved [56,57].

On the other side, the important dispersion observed on the global scores provided by the panel of experts highlights another relevant aspect which is the diverse criteria used, considering the experience and personal judgment on performance. Therefore, further work should be done on aligning and defining common and objective criteria that support prehospital trauma management evaluation.

This study has several limitations. First, individual training is considered in which a single part of the body is affected. Nevertheless, when there is a severe trauma lesion, more than one body part is affected. Second, this study considered the sequence of actions performed by the students to evaluate trauma management performance. Finally, more students should be enrolled to confirm the preliminary findings obtained within this work.

### Future Directions

Future studies should work on connecting the web-based trauma simulator to physical simulators: low, medium, or high-fidelity simulators. This will allow practicing, additionally, technical skills and procedures that are needed to stabilize a trauma patient. Moreover, different profiles should be included in the simulator, which will allow customizing training depending on the target audience. Additionally, more than one body part could be affected by trauma; therefore, this should be implemented within the web-based simulator. This will allow practicing polytrauma management and, together with the implementation of new profiles, will allow us to interact with different profiles needed when treating a trauma patient. Finally, a more comprehensive score set could be developed for the evaluation of the trauma management scenarios. This will allow the experts to provide more than one score, which may give more information about the training gaps in undergraduate prehospital trauma management.

## 5. Conclusions

Our study shows that there is a need to train prehospital trauma management in medical education and that an objective evaluation criterion should be developed. However, this is an initial investigation, and more research should be carried out. This will improve trauma management knowledge acquisition and retention, with a focus on transferring it to real prehospital scenarios.

## Figures and Tables

**Figure 1 ijerph-19-13188-f001:**
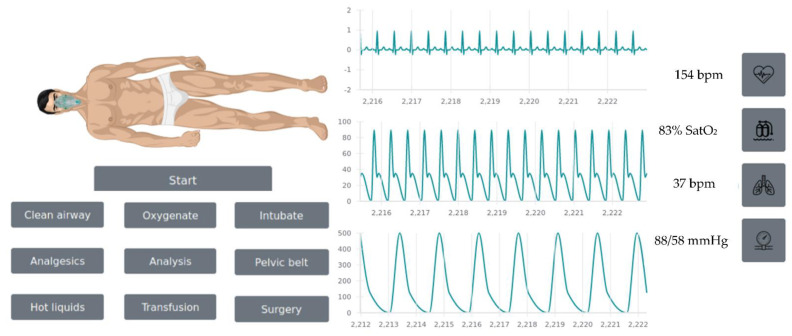
Web-based simulator in which the virtual patient is shown together with his vital signs and some of the actions that could be accomplished.

**Figure 2 ijerph-19-13188-f002:**
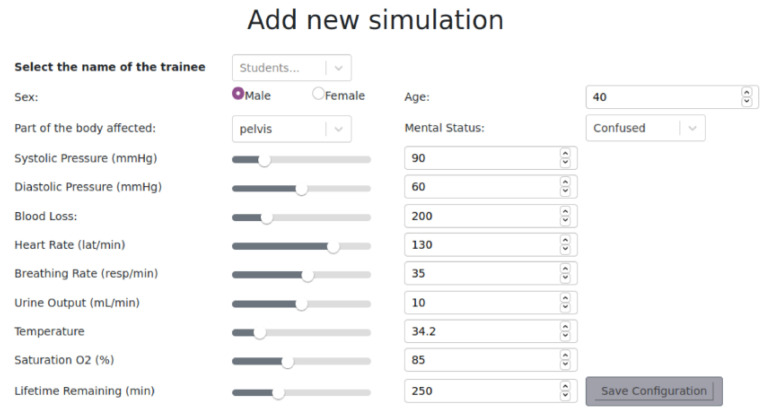
Trauma scenario definition screen. This section is only available for trainers in which the trauma scenario, together with the trainee, will be defined and selected.

**Figure 3 ijerph-19-13188-f003:**
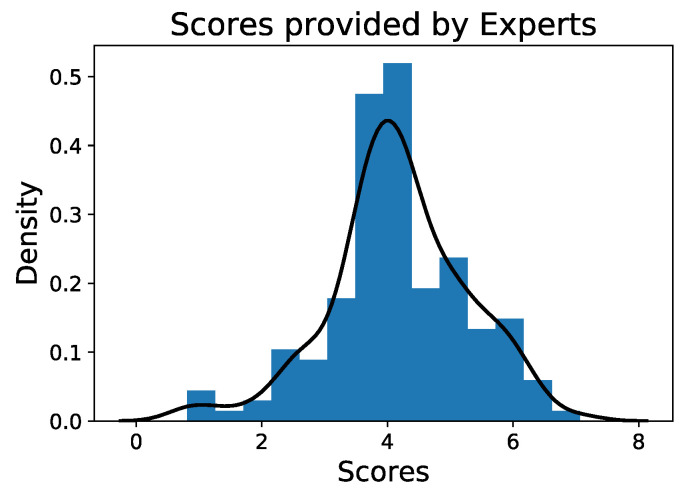
Histogram of all the average scores provided by the panel of experts regarding 78 simulations created.

**Figure 4 ijerph-19-13188-f004:**
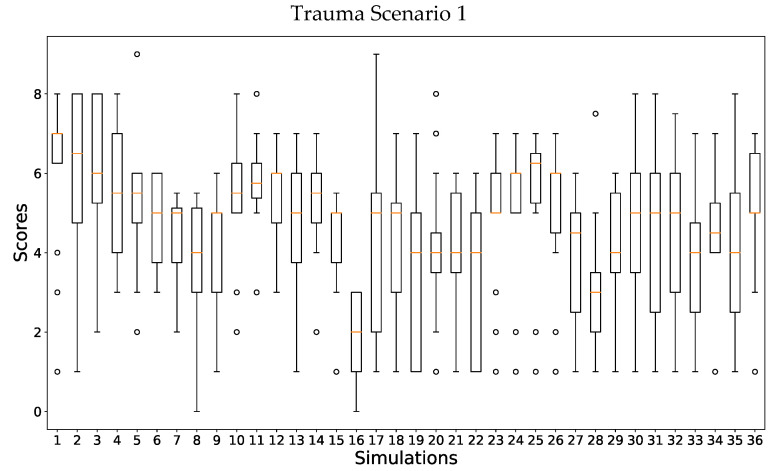
Scores provided by the panel of experts to all simulations performed for the trauma Scenario 1: pelvic trauma scenario.

**Figure 5 ijerph-19-13188-f005:**
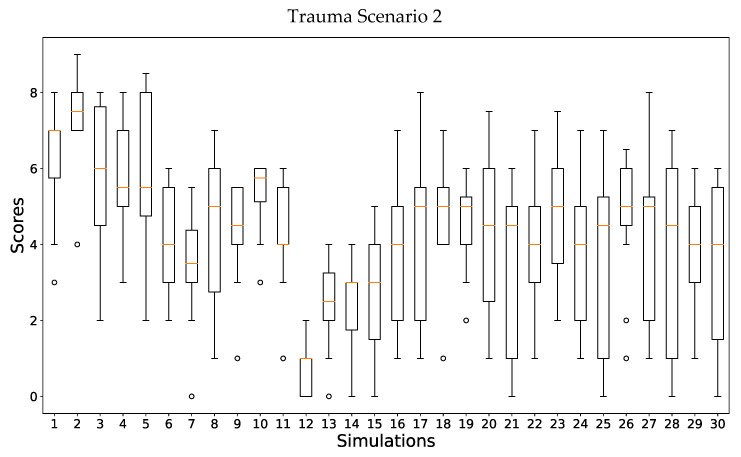
Scores provided by the panel of experts to all simulations performed for the trauma Scenario 2: lower limb trauma scenario.

**Table 1 ijerph-19-13188-t001:** Scores provided by the trauma experts to all the simulations performed for the two trauma scenarios trained (mean ± SD). At the end of the table, in bold, the mean and standard deviations of the scores per trauma scenario are shown.

Trauma Scenario 1	Trauma Scenario 3
6.12 ± 1.93	6.25 ± 1.55
6.00 ± 2.01	7.06 ± 1.54
5.87 ± 2.15	5.75 ± 2.10
5.44 ± 1.64	5.75 ± 1.37
5.69 ± 1.94	5.81 ± 2.01
4.81 ± 1.18	4.25 ± 1.46
4.31 ± 1.18	3.56 ± 1.48
3.62 ± 1.59	4.31 ± 1.96
4.19 ± 1.63	4.44 ± 1.18
5.37 ± 1.55	5.19 ± 1.16
5.81 ± 1.44	4.31 ± 1.24
5.44 ± 1.22	0.81 ± 0.63
4.87 ± 1.69	2.50 ± 1.17
5.12 ± 1.43	2.56 ± 1.17
4.06 ± 1.59	2.67 ± 1.49
2.00 ± 0.91	3.80 ± 1.72
4.40 ± 2.24	4.30 ± 2.29
4.27 ± 1.72	4.47 ± 1.93
3.60 ± 2.12	4.60 ± 1.02
4.13 ± 1.77	4.37 ± 2.17
4.20 ± 1.55	3.47 ± 2.12
3.47 ± 1.93	4.13 ± 1.75
4.93 ± 1.65	4.80 ± 1.79
5.07 ± 1.92	3.67 ± 1.84
5.25 ± 2.11	3.30 ± 2.37
5.10 ± 1.86	4.83 ± 1.50
3.87 ± 1.77	4.17 ± 2.00
3.10 ± 1.53	3.80 ± 2.45
4.13 ± 1.66	3.73 ± 1.56
4.67 ± 1.81	3.53 ± 2.12
4.33 ± 1.92	
4.47 ± 1.83	
3.73 ± 1.69	
4.20 ± 1.87	
4.13 ± 1.99	
5.17 ± 1.58	
**4.58** **± 1.71**	**4.21** **± 1.67**

## Data Availability

Data available on request.

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
