# Peer review of "The Need for Trauma Management Training and Evaluation on a Prehospital Setting"

_ijerph, 2022, doi:10.3390/ijerph192013188_

Round 1
Reviewer 1 Report
Dear authors - thank you for the opportunity to review the article. The topic of training in prehospital trauma care in medical studies and continuing medical education is highly relevant. I fully agree with you that there should be a stronger focus here in the future.
The biggest weakness in my eyes (and this is already addressed by you) is the lack of standardisation of the assessment. In my opinion, the statement that the students/residents are poorly trained cannot be formally made in the end. Because it would be possible that with a different horizon of expectations, they would all be assessed with 8 or 9. Especially in the field of assessment, standardisation is very important to create comparability. Checklists also offer a good opportunity in the context of workplace-based assessments.
I have a few comments regarding your brief report:
- In line 34 you write that ATLS courses are not open to medical students. I think the rest of the manuscript talks about a few places. This should be standardised. In my country, participation by medical students is possible - although rather rare because of the course fees.
- In the Materials & Methods section it would be very useful for the reader to see some screenshots of the simulator to have a rough overview. Possibly, further material could also be added in the supplement. I found a short video in Spanish on the GitHub homepage.
- For the understanding of the simulator, it would also be good if perhaps the screenshots were used to explain the principle in more detail. What role does the examinee take on? The role of the trauma leader? Polytrauma care is basically done in a team - how is that taken into account in the simulator? Because different experts normally come together here (trauma surgery, emergency medicine, anaesthesiology/intensive care, etc.). Is the simulator a linear query of facts? Or are there multidimensional decisions?
- An institutional review board does not appear to have been consulted. Is there a consent form or information on the anonymity/data protection of the participants?
- Was the candidate known to the examiners/evaluators? Whether it was a medical student or resident?
- Was there any form of training for the examiners or the creation of a target profile for 0 or 10? Was there a checklist or something similar? Which subject area do the examiners come from? What exactly defined them as an "expert"?
- In the footnote of Figure 2, there is one point too many at the end.
- As already described, in my view the statement in the discussion cannot be made in this way due to the lack of standardisation of the examiners. It would be nice if reference could be made again to workplace-based examination formats (EPAs, multisource feedback, etc.).
- It might also be useful to mention the possibility of team-based simulation training on high fidelity simulators. At least in my country, this is also trained for trauma care in medical studies at some universities (although this alone is of course still not sufficient).
- Another aspect for discussion would be whether apps for smartphones/tablets that have the relevant guidelines stored can support the care of polytrauma patients - especially for inexperienced centres or colleagues.
I wish you all the best for the revision of the manuscript!
Reviewer 2 Report
The article is very interesting since it deals with different aspects of learning in trauma care at the university level. However, I think that, to be published, it has to improve many aspects in terms of the description of the study they carry out, but above all the results they show us. If the authors improved the results, I think they could go much deeper into the discussion that they show us that I see as a bit poor.
Introduction.
I think they should focus a little more on the current teaching of trauma management in the university setting and how to teach it.
I think that the authors should write a paragraph in relation to clinical simulation in university learning and in the field of specialized training, as well as the evaluations that can be made of it.
In general the paragraphs are too long I think they should be shortened somehow.
The authors should write in the last paragraph of the introduction the main objective of the study since it is not clear at all.
Material and methods.
The authors should specify the form of selection of the students and residents if they were chosen randomly or were volunteers. They should also specify whether these simulations were done individually or in teams and how many people participated.
What specialties are the experts from and what were the selection criteria? The evaluation was carried out while the student was performing the simulation or it was done after watching the recording or only the objective actions that the student carried out were analyzed.
The authors would have to give us what aspects of each student they evaluate.
The authors would have to write the statistical analyzes they perform.
Results.
In general, I believe that the results can be greatly improved from the point of view of the description of the findings they make and go much deeper into the numerical data they provide us.
Authors should not draw conclusions in this part of the article as they do in paragraph 1.
The authors should have analyzed the grades obtained depending on whether they are students or residents, as well as an assessment of the grades obtained in each of the aspects that the experts have evaluated and if there have been differences in terms of scanning them. I think that the analysis they make of the grades obtained by the students is very poor.
The authors would have to give the data of means or medians with dispersion results. (standard deviation or range) depending on the data provided to us.
The results they provide on the differences in the scores of the experts are difficult to understand and very improvable.
The graphs that they show us contribute little, I think that they would contribute some table in which the results and the differences in the evaluation of the students in numerical form would be grouped.
Discussion
In general, I observe that the discussion is a bit poor, and the authors only repeat the results obtained to us without delving into them or comparing their results with any other study carried out, they could compare with other studies on simulation.
Has there been a study similar to the one the authors have proposed in which they can compare their results? Even if it is in other health fields? Or even with other pathologies to be able to assess the problem of dispersion in the notes of the experts or even in the difficulty for the evaluation in these educational environments.
IF the authors analyzed which aspects of the student are more deficient, I think they could delve deeper into the aspects of improvement in the education and training of students.
Round 2
Reviewer 1 Report
Thank you very much for the detailed revision of the manuscript. In my opinion, the additions and clarifications have once again significantly improved the article.
Author Response
Dear reviewer, we thank your interest shown on the manuscript and your suggestions and comments that improved this document.
Thank you very much.
Reviewer 2 Report
I would like to congratulate the authors for the improvements made to the text. The authors have answered most of my questions, but there are still some aspects that should be clarified or modified.
Introduction.
I think that in the objective they should reflect that the study is carried out among medical students.
Material and methods.
The authors do not specify in their explanation how the students who participated in the study were chosen.
I think that the answer they give us does not answer the question, since what they show us again is the objective of the study. I do not observe in the text the form of selection of the students. I think the answer they offer me does not answer the question that arises.
Results.
The authors should show us in paragraph 1 the standard deviation and the interquartile range that they obtain by showing us the mean and the median, respectively.
The authors again show us some conclusions, at the end of paragraph 1 without having modified it, I ask you to review it.
The authors refer to a comparison between the means obtained by the two scenarios that they have proposed, they should indicate in the material and methods that statistical tests have been used.
The authors have not analyzed the average marks of each item that the experts have evaluated in each scenario, I believe that this would help to assess the training deficiencies that the students could have.
Discussion.
In the first paragraph they show us some results that should have been shown in the results section. I do not understand where they get this data from and it is very confusing to me, since they do not coincide with those shown in table 1. I do not know if they refer to the same results.
One aspect that I do not observe and that I think is fundamental is that the authors should show us the limitations of the work.
